# A social image recommendation system based on deep reinforcement learning

**Somaye Ahmadkhani**[iD]**, Mohsen Ebrahimi Moghaddam**[iD]*

Shahid Beheshti University, Faculty of Computer Science and Engineering, Tehran, Iran

* m_moghadam@sbu.ac.ir

## Abstract

Today, due to the expansion of the Internet and social networks, people are faced with a vast amount of dynamic information. To mitigate the issue of information overload, recommender systems have become pivotal by analyzing users' activity histories to discern their interests and preferences. However, most available social image recommender systems utilize a static strategy, meaning they do not adapt to changes in user preferences. To overcome this challenge, our paper introduces a dynamic image recommender system that leverages a deep reinforcement learning (DRL) framework, enriched with a novel set of features including emotion, style, and personality. These features, uncommon in existing systems, are instrumental in crafting a user's characteristic vector, offering a personalized recommendation experience. Additionally, we overcome the challenge of state representation definition in reinforcement learning by introducing a new state representation. The experimental results show that our proposed method, compared to some related works, significantly improves Recall@k and Precision@k by approximately 7%–10% (for the top 100 images recommended) for personalized image recommendation.

## 1 Introduction

In recent years, with the development of the Internet and social networks, impressive research effort has been directed at recommender systems. This is because, in the face of the huge volume and variety of information, there is a need for a system that can automatically identify and cater to the user's interests. Despite various advances in recommender systems, existing ones still require further improvements to provide more efficient recommendations that are applicable to a broader range of applications. The exponential increase in images as an important source of information in online services and social networks has prompted us to investigate the image-based recommender systems of social networks.

Prior studies have explored handcrafted features [1, 2], as well as deep learning models [3–5] for image representation and recommendation. Feature selection and the use of deep learning models have been prominent in many studies [6–8]. In recommender systems, various similarity measures [9] and classification methods [10] have been employed. A remarkable point in all of this is that the recommendations are often viewed as a static procedure, assuming that the user's underlying preferences remain unchanged. In reality, personalizing image

**Data Availability Statement:** We have described the dataset and its location in detail on our GitHub repository, which contains the minimal data set underlying the results described in our manuscript.

You can find this dataset and related materials at the following GitHub link: https://github.com/Samadkhani/ImageRecommendation/tree/main Our GitHub repository includes comprehensive documentation and access instructions to ensure that the minimal data set is readily available to replicate the study's findings in their entirety.

**Funding:** The author(s) received no specific funding for this work.

**Competing interests:** The authors have declared that no competing interests exist.

recommendations according to individual user preferences is a complex task. User preferences evolve over time, necessitating a dynamic interaction between the system and the user, thereby indicating the need for a dynamic recommendation process. Additionally, when data is limited, making accurate recommendations becomes challenging.

Reinforcement learning (RL) is increasingly used for its ability to tackle complex problems involving dynamic modeling and long-term planning [11]. The use of RL in recommender systems is not new and has been used in earlier works. Model-based RL techniques such as Partially Observable Markov Decision Processes (POMDP) [12] and Q-learning [13] were among the first RL techniques applied to modeling recommendation methods. However, these methods can become inefficient when the number of proposed items is large, due to their time complexity. Later on, model-free RL techniques were also employed for recommendations. These can be categorized into two groups: value-based [14–16] and policy-based [16, 17].

Although the use of reinforcement learning (RL) in recommender systems is not new, scalability issues rendered traditional RL algorithms impractical for widespread use. With the advent of deep RL, a new trend has emerged in this field, enabling the application of RL to recommendation problems with large state and action spaces[18].

For example, the authors in [19] propose a recommendation framework based on deep RL. They model the interactions between the users and recommender systems using an Actor-Critic RL scheme and consider both dynamic adaptation and long-term rewards. Zhao et al. [16] employed deep RL to automatically learn optimal recommendation strategies and model the recommendation as a Markov decision process.

Due to the effectiveness of recommender systems based on Reinforcement Learning (RL) in addressing real-world problems, we propose the use of deep RL to delineate the sequential interactions between users and the recommender system. This approach facilitates the automatic learning of optimal strategies from user feedback.

Our proposed method comprises two main parts with following innovations in each one. In the first part, we aim to extract a diverse set of features from images to capture the characteristics and preferences of the user. For this purpose, we introduce three novel components: emotion analysis, personality recognition, and style detection. These elements are designed to represent users more accurately and enhance the social image recommender system. In the second part, we introduce a new framework based on actor-critic RL, complemented by a state representation module, to establish a dynamic recommender system.

The main contributions of this study can be summarized as follows:

1. Proposing an image recommender system in a new deep RL framework. Unlike traditional methods that treat recommendation as a static process, our methodology acknowledges the dynamic nature of user preferences. By utilizing a deep RL framework, our system not only reacts to user feedback in real-time but also predicts future preferences. This approach improves the system's efficiency in making recommendations with limited data.

2. Introducing a new method for state representation. We applied the concept of self-attention, a core mechanism of the Transformer, to compute the new state in our system. This method focuses on measuring dependencies within a sequence's components, offering a more nuanced understanding of user preferences.

3. Introducing three components (style, Emotion, personality) that can be useful in images recommendation to create user's characteristics vector and investigate its effect on the recommended images on each user. These features are not commonly used in current systems but have shown to significantly impact the system's ability to make personalized recommendations, as demonstrated in our test results.

In summary, our methodology introduces significant improvements by adopting a dynamic perspective on user preferences, integrating novel features for personalization, and leveraging the latest advances in deep learning and reinforcement learning. These enhancements not only address the challenges faced by existing systems but also pave the way for future research in the field.

The structure of this paper is as follows: related works and background are presented in Section 2. The proposed methods are introduced in Section 3. Experimental details and results are discussed in Section 4. Finally, we conclude this paper in Section 5 and discuss some future work.

## 2 Image recommendation related works

Recommender systems have garnered significant attention from researchers and are utilized across various fields [20–24]. In this section, we divide recommender systems into two categories: Non RL or traditional recommendation systems and RL based recommendation systems. Traditional recommender systems can be divided into three categories: collaborative filtering, content-based filtering, and hybrid [25]. Collaborative filtering recommendation are based on rates given to the items by the users and using rates to find similar users. Content-based filtering is based on content features of items to find similarities between items. Hybrid methods use the capabilities of both methods. With the increasing number of images on the Internet and social networks, image recommender systems have become a critical area of research and have garnered considerable attention. Therefore, our focus is on image-based recommendations, especially those involving social images.

Some studies have investigated feature extraction methods from images to understand user interests. Lovato et al. [1] considered the images tagged as favorites by a user and extracted hand-crafted features from them such as HSV statistics, Use of light, and etc. Although these features can be useful, there is still a need to extract more features. Guntuku et al. [10], in addition to low-level features [1], extracted a set of high-level features such as Head and Upperbody recognition, Visual Clutter, tag, etc. They proposed a deep bimodal knowledge representation model that increased efficiency by using visual and tag features. The method presented in [26] extracts various features, applies feature selection to take the important features, and then uses fuzzy inference system for image recommendation. They assert that fuzzy logic, as a decision-making system, adeptly handles vague and inaccurate information. This makes it suitable for modeling the uncertain and ambiguous preferences of users in recommender systems. In the methods described, image-level features and social information, such as tags, are utilized to enhance recommendations. In [27], introduction of style as a new component in social image recommendation is presented. This demonstrates that incorporating style significantly improves the results.

In some previous researches, image-level features, deep methods, and social information have been noticed to determine user preferences [28–30]. However, defining a new component related to user preferences can be useful.

Most traditional recommendation methods involve user-item interaction modeling with supervised learning such as classification, memory-based content filtering from user history, etc. These methods ignore the dependence during successive time steps. Unlike the supervised learning setting, where a guide tells you the right action, to better reflect the user-system interaction, it is widely agreed that the formulation of the problem as a sequential decision problem can be better [18]. Therefore, it can be solved by RL relies on the environment to discover the right action. In RL, the learning process is through interaction with an environment.

Shani et al. [12] used Markov decision processes (MDPs) for recommender systems to consider the long-term effect of recommendations and the expected value of each

recommendation. Taghipour et al. [13] used Q-Learning to model web page recommendation. However, these model-based RL techniques [13] are inapplicable when there are many candidate items for recommendation, because updating the model requires a time-consuming dynamic programming step. Therefore, model-free RL techniques were preferred for use in recommender systems. These techniques can be divided into two categories: value-based [14, 15] and policy-based [17, 18].

In value-based approaches, for a given state, one must calculate the Q values of all available actions, then the action that has the maximum Q-value is selected as the best action. Therefore, when we face a very large action space, the approaches may become very inefficient.

The policy-based approaches generate a continuous parameter vector to represent an action [17, 18]. This vector can generate the recommendation and update the Q-value evaluator; so can overcome the inefficiency drawbacks.

As deep neural networks developed; deep RL emerged. Deep RL techniques have recently attracted the attention of recommender systems because they enable the use of RL in problems with large state and action spaces. Zheng et al. [14] applied deep Q-learning in news recommendations to effectively model the dynamic news features and user preferences. The Dueling Q-network is utilized in [14] to model the Q-value of a state-action pair. These value-based approaches require evaluating the Q-values of all actions under a specific state, which becomes highly inefficient when the number of actions is large. Zhao et al.[16] proposed List-wise recommendations based on Deep RL, which can be applied in scenarios with large and dynamic item space. In another study, Zhao et al. [31] used Deep RL for page-wise recommendation to learn the optimal recommendation strategies and optimizes a page of items simultaneously. However, there was another limitation concerning the user state, which did not explicitly and carefully model the interactions between users and items. Liu et al. [19] used an actor-critic RL approach to model the interactions between users and recommender systems for dynamic adaptation and long-term rewards. Huang et al. [32] considered the recommendation process as a Markov decision process and proposed a top-N model based on deep RL for long-term prediction, wherein recurrent neural network is used to simulate the interactions between the recommender system and users.

## 3 Proposed method

Fig 1 shows the workflow of the proposed method, which consists of two main parts: feature extraction and deep RL for learning the recommendation process. For feature extraction, we propose the use of three new components: emotion analysis, personality recognition, and style detection, aiming to enhance the social image recommender system. In the second part, we propose a new framework based on actor-critic RL to develop a dynamic recommender system. In the following sections, we will first describe the methods used for feature extraction, and then we will present the proposed recommender system, which is based on deep RL and the extracted features.

### 3.1) Feature extraction

In our proposed recommender system framework, four key components contribute to feature extraction as shown in Fig 1. These components are: Visual Feature Extractor, Style Feature Extractor, Emotion Feature Extractor, and Personality Feature Extractor. Each of these features is listed in Table 1. Below, we describe the model designed and used for each component. For every image, we extract an emotion vector to express the image's emotions, a style vector to denote the most prominent styles, and a personality vector to represent key personality traits. Our interpretation hinges on the concept that a user tags an image as a favorite when it

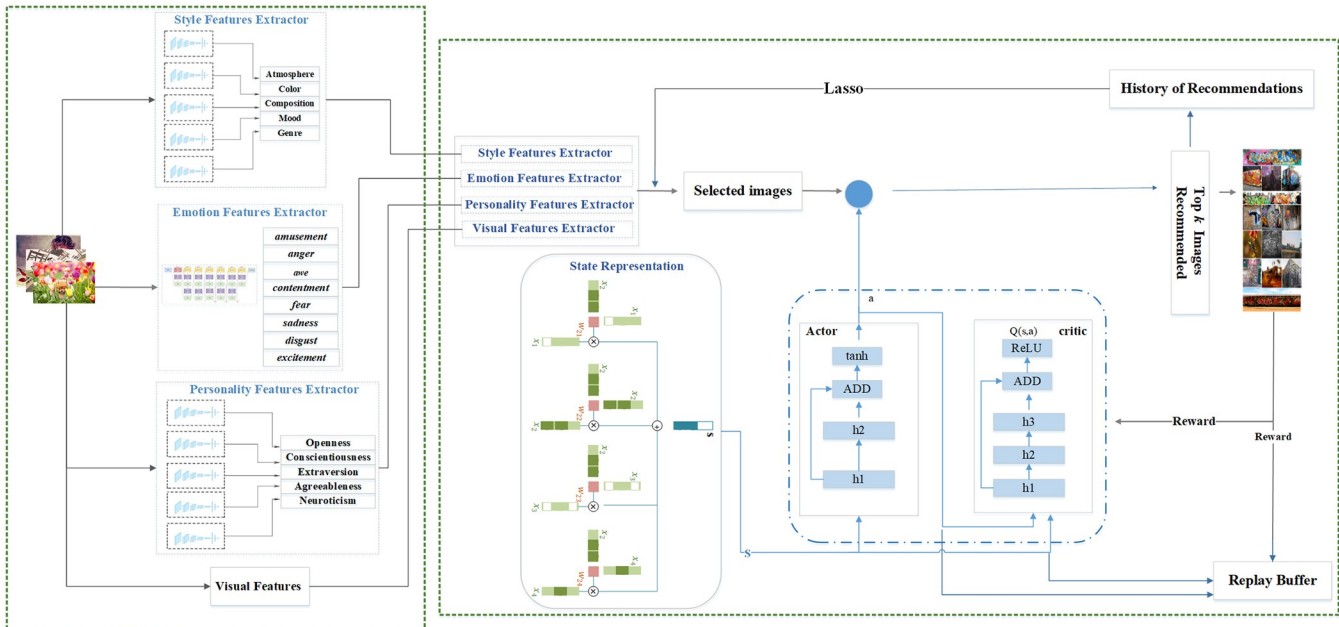

**Fig 1. Workflow of the proposed method (The images shown as input and output in this workflow are from reference [1]).**

aligns with their interests in terms of emotion, personality, and style. To extract these three distinct features, we initially train a separate model for each. We then produce the corresponding feature vector by inputting an image into each model. Upon extracting these components, we utilize RL to introduce a novel social image recommender system.

**A) Emotion feature extractor**. Understanding emotions in images has attracted much interest due to its various applications. Many researchers, inspired by psychology and artistic principles, have investigated various features extracted from images to automatically assign a single emotion to each image [33, 34]. In the past few years, with the popularity of convolutional neural networks (CNNs), researchers [35, 36] have utilized CNNs to recognize image sentiments and have demonstrated that deep features outperform hand-tuned features. Recent algorithms in convolutional neural networks have significantly improved emotion classification, which aims to detect differences between emotion categories and assign a predominant label to each image. Some studies have focused on increasing the depth of neural networks, and in some cases, it has been shown that as the depth of the network increases, the ability of the network to extract higher-level features improves. The EfficientNet neural network [37] states that CNN models should be scaled meaningfully to achieve better accuracy and efficiency. The authors of [37] propose that carefully balancing the network's depth, width, and resolution can improve performance. Based on this, a new scaling method has been proposed that scales all dimensions of depth, width, and resolution uniformly using a simple yet very effective combination factor.

In the present paper, we investigate the utilization of a new deep neural network, Efficient-Net, which is expected to yield better results. This method involves expanding the network not just in one dimension—depth, width, or resolution—but in a combined manner to achieve maximum efficiency with fewer parameters.

The integration of emotion as a component in our recommender system represents a novel aspect of our research. We employ the fine-tuned EfficientNet network to analyze the emotions conveyed by the images.

**Table 1. List of features.**

|  | Category | Name | Feature Dimensions |
|---|---|---|---|
| **Visual Feature** |  | **Use of light** | 1 |
|  |  | **HSV statistics** | 3 |
|  |  | **Emotion based** | 3 |
|  |  | **Hue Circular Variance** | 1 |
|  |  | **Colorfulness** | 1 |
|  |  | **Color name** | 11 |
|  |  | **Entropy** | 1 |
|  |  | **Wavelet textures** | 12 |
|  |  | **Tamura** | 3 |
|  |  | **GCLM features** | 12 |
|  |  | **Edges** | 1 |
|  |  | **Level of detail** | 1 |
|  |  | **Regions** | 1 |
|  |  | **Low depth of field (DOF)** | 3 |
|  |  | **Faces** | 1 |
|  |  | **GIST descriptors** | 24 |
| **Style Feature** | Atmosphere | **Hazy** | 1 |
|  |  | **Sunny** | 1 |
|  | Color | **Bright** | 1 |
|  |  | **Pastal** | 1 |
|  | Composition | **Detailed** | 1 |
|  |  | **Geometric** | 1 |
|  |  | **Minimal** | 1 |
|  |  | **Texture** | 1 |
|  | Mood | **Serene** | 1 |
|  |  | **Melancholy** | 1 |
|  |  | **Ethereal** | 1 |
|  | Genre | **Noir** | 1 |
|  |  | **Vintage** | 1 |
|  |  | **Romantic** | 1 |
|  |  | **Horror** | 1 |
| **Emotion Feature** |  | **Sadness** | 1 |
|  |  | **Fear** | 1 |
|  |  | **Excitement** | 1 |
|  |  | **Disgust** | 1 |
|  |  | **Contentment** | 1 |
|  |  | **Awe** | 1 |
|  |  | **Anger** | 1 |
|  |  | **Amusement** | 1 |
| **Personality Feature** |  | **Openness** | 1 |
|  |  | **Conscientiousness** | 1 |
|  |  | **Extraversion** | 1 |
|  |  | **Neuroticism** | 1 |
|  |  | **Agreeableness** | 1 |

**B) Personality feature extractor.** Personality analysis can be one of the most important methods for distinguishing user preferences and behaviors. To date, many studies have investigated personality analysis [38, 39]. In our research, we have examined personality analysis as one of the influential components in the social image recommender. For predicting personality traits, we utilize pre-trained CNNs designed for image classification. These networks are trained using a large set of images, with intermediate layers capturing the semantics of general visual appearance. We harness the capability of these networks, fine-tuning them for our specific problem to learn visual representations correlated with personality traits.

Our approach is similar to the one implemented in [40], where the study models users' personality traits based on the Five Factor Theory. Therefore, five distinct binary classifications are considered for each trait. We have considered CNN networks, trained on the ImageNet, fine-tuning the network, and changed the last layer to adapt binary classifications for each trait.

**C) Style feature extractor.** Image style plays an important role in how the image looks. Automatic image style recognition is crucial for many applications, including artwork analysis, photo organization, and image retrieval [41–43]. However, its use in social image recommender systems has not been widely explored. This study proposes image style detection as a component to better understand user preferences. For this purpose, we first obtain the style vector for each image. Our previous work [27] proposed an image style detection method based on deep correlation features and a compact convolutional transformer (CCT). This method is based on convolution and tries to preserve local information. The idea in this method is that a new convolutional block is proposed instead of the simple convolutional block in CCT. For this purpose, we use VGG-19 pre-trained convolutional layers, which are trained using the ImageNet dataset as a convolutional block, and fine-tuned part of it during the compression transform learning process. In this paper, we have used the proposed method in [27] to extract the style features of images.

**D) Visual feature.** Visual features listed in Table 1 are part of the features used in [1]. This list represents a wide, albeit not exhaustive, spectrum that is useful for our purpose. It takes into account cues focusing on aesthetic aspects [34, 44], and includes focus on faces, as well as the adoption of GIST scene descriptors [45], which involve applying a set of oriented band-pass filters. Some of these features are explained below. For more complete information, refer to [1].

The **use of light** is a fundamental property of image aesthetics. Underexposed or overexposed pictures are usually considered to be of poor quality. In the HSV color space, the lightness of an image is measured as the average pixel value of the V channel. **HSV statistics**, specifically the mean of the S channel and the standard deviation of the S and V channels, are also analyzed. **Emotion-based** Saturation and Brightness directly influence pleasure, arousal, and dominance. The equations used for computing these are as follows: Pleasure = 0.69V + 0.22S, Arousal = -0.31V + 0.60S, and Dominance = 0.76V + 0.32S. **Colorfulness** allows us to distinguish between multi-colored images and those that are monochromatic, sepia, or simply low in contrast. It is measured using the Earth Mover's Distance (EMD) between the image's histogram and a flat histogram, which represents a uniform color distribution. This measurement follows the algorithm suggested by Datta et al. [44]. **Color naming** is a photographer technique often use to showcase their personal style. Following [1], we consider the following 11 color names: black, blue, brown, grey, green, orange, pink, purple, red, white, and yellow.

## 3.2) Recommendation based on RL

In this research, we construct a user recommendation model for each individual user, denoted as user $i$, based on a collection of M images that the user has marked as favorites. Imagining a

scenario with N distinct users, represented as $U = \{u_1, u_2, \ldots, u_N\}$, and a corresponding set of M images $I = \{I_1, I_2, \ldots, I_M\}$, with $I_i = \{x_1^i, x_2^i, \ldots, x_k^i\}$ signifying the subset of images liked by user $I$, and $e_i$ represents the feature vector representation of item $x^i$. Accordingly, a portion of each user's preferred images is allocated for the training set, while the remainder is utilized for testing purposes.

We model our proposed method in the RL setting. An agent is a recommendation system in our context, which interacts with the environment (users) and receives rewards from the environment (feedback from users). Rewards serve as an index of whether the course of action the agent is taking is right or wrong. The agent is constantly in contact with the environment, and eventually learns to take the right action through the feedback received from the environment over a period of time. The underlying RL model is the Markov Decision Process (MDP) includes a sequence of states, actions, and rewards. MDP is determined by five components as follows:

**State space $S$**: Created by the environment, this represents the agent's situation at any given moment. In our work, the status s indicates the user's positive interaction history with the recommender, and we have proposed a state representation module to represent it (Section 3-2-s1).

**Action space $A$**: An action $a \in A$ includes possible reactions that the agent may exhibit in response to the current state. This could be a list of items recommended to the user in the current state.

**Reward $R$**: This is the immediate feedback sent to the agent by the environment after each action is evaluated.

**Transition probability $P$**: This denotes the probability of transitioning from state $s_t$ to state $s_{t+1}$ when the agent performs action $a_t$.

**Discount factor $\gamma$**: It is a factor that measures the present value of long-term rewards. Its value is between 0 and 1. If $\gamma = 0$, the recommender ignores long-term rewards and considers only immediate ones. However, when $\gamma = 1$, the recommender considers long-term and immediate rewards equally important.

As previously mentioned, most work has focused solely on static recommendation processes using pre-trained models, which do not effectively simulate the dynamic interaction between users and their systems. Therefore, we propose a new social image recommender system within the deep RL framework. For this purpose, the deep deterministic policy gradient (DDPG) algorithm has been employed to generate recommendations. DDPG is well-suited for RL tasks that involve continuous action spaces and complex, high-dimensional state spaces, and it can also simultaneously reduce redundant computation. These characteristics make it ideal for recommender systems. DDPG is an actor-critic method that merges the strategies of Q-learning and policy gradients, consisting of two models: the Actor network and the Critic network.

**The actor, or policy network,** takes the user's state as input and generates an appropriate action. We determine the user's state vector at each step using the state representation module, which assesses the user's state based on their history of liked images (The proposed module for state representation is described in Section 3-2-1). The actor-network is composed of two ReLU layers, one Tanh layer, and incorporates a skip connection in the penultimate layer. This network receives the user state and outputs the action vector, which is then used to make recommendations to the user. Since there are numerous images to potentially recommend, we employ a scoring function based on the inner product, as follows, to select the most

appropriate image for recommendation.

$$score_i = e_i^T a \tag{1}$$

Where $a$ is the output of the Actor and $e_i$ expression is the vector representation of item $x^i$. After calculating the scores, the images are ranked based on these scores, and the image with the top rank is recommended.

**The critic network** is a Deep Q-Network designed as a deep neural network parameterized by $Q(s, a)$. The generated action vector and the user's state are given as input to the critic network to determine how good the generated action is. According to $Q(s, a)$, the Actor network's parameters are updated in order to improve performance of action $a$, that is, to enhance $Q(s, a)$.

In the Deep Deterministic Policy Gradient (DDPG) algorithm, both the actor and critic networks have their respective loss functions, which guide the learning process. These loss functions are central to the training and updating mechanisms. The critic is updated, with the objective of minimizing the difference between the predicted Q-values and the target Q-values. The loss function for the critic is typically the Mean Squared Error (MSE) between these two values:

$$L(\theta^Q) = \mathbb{E}_{(s,a,r,s') \sim \mathcal{D}} \left[ \left( Q(s, a|\theta^Q) - y \right)^2 \right] \tag{2}$$

Where $\theta^Q$ are the parameters of the critic network, and $y$ is the target value, computed as $y = r + \gamma Q'(s', \mu'(s'|\theta^{\mu'})|\theta^{Q'})$. Here, $\gamma$ is the discount factor. $Q'(s', \mu'(s'|\theta^{\mu'})|\theta^{Q'})$ is the target Q-value for the next state, using the target critic network's parameters, $\theta^{Q'}$, and the target actor network, $\mu'$, for action selection. $\mathcal{D}$ represents a replay buffer containing tuples of (state, action, reward, next state).

The policy of the actor is updated using policy gradient methods. The objective is to maximize the Q-value, output by the critic, for the actions chosen by the actor. However, in practice, gradient ascent is performed on the expected Q-values predicted by the critic for the current policy. The loss function for the actor essentially becomes the negative of the expected Q-value:

$$L(\theta^\mu) = -\mathbb{E}_{s \sim \mathcal{D}} \left[ Q(s, \mu(s|\theta^\mu)|\theta^Q) \right] \tag{3}$$

Where $\theta^\mu$ are the parameters of the actor network. $\mu(s, \theta^\mu)$ is the current policy's action given state $s$. The expectation is approximated by sampling from the replay buffer.

For each iteration (or batch of experiences), the critic's parameters $\theta^Q$ are updated by minimizing the critic loss. This typically involves calculating gradients of $L(\theta^Q)$ with respect to $\theta^Q$ and then performing a step of gradient descent. The actor's parameters $\theta^\mu$ are updated by performing gradient ascent on the expected Q-values. The gradients required for the update are derived from the critic's Q-value output with respect to the actor's parameters. This is where the dependency of the actor on the critic's assessment comes into play, guiding the actor to choose actions that the critic values more. These updates are typically applied in a loop, where a batch of experiences is sampled from the replay buffer, the critic is updated, and then the actor is updated. Target networks for both the actor and the critic are also maintained, with their weights being slowly tracked to the main networks' weights to stabilize learning. This process repeats for many episodes and steps, gradually improving both the actor's policy and the critic's value estimation.

Given the vast number of images available for recommendation, we propose utilizing a memory of the user's recommendation history to aid the recommendation process with the help of the Action vector. This memory records all images recommended to the user, including

those they liked and those they did not. We employ the Lasso algorithm to derive a vector of coefficients from the stored information. We then calculate the dot product between this coefficient vector ($\beta$) and the candidate images to select a list that represents the most suitable recommendations.

$$score_{i\beta} = x_j\beta \tag{4}$$

After calculating the $score_{j\beta}$ for all $x_j$, we consider those with higher scores as better candidates for recommendation. We then compute the dot product of the list of selected images with the action vector obtained from the actor network to create recommendations, as outlined in Eq (1). The reward function is defined in Eq (5). If a recommended image is liked by the user, the system receives a positive reward; conversely, if the image is not liked, the system receives a negative reward.

$$Reward(Image) = \begin{cases} 1 & \text{If the user likes the image} \\ -1 & \text{If the user does not like the image} \end{cases} \tag{5}$$

**3.2.1) state representation module.** State representation is a challenging problem in RL. To address this, we have proposed a new method for representing the user's state, employing the self-attention concept [46]. This approach operates in a manner where, at every step, when an image is recommended to the user, and if the user likes it, we capture the self-attention of the recommended images alongside other images in the collection that the user likes. Eq (6) shows how to update the user state. We represent the current status as $s_c$, the next state as $s_N$, and 'PositiveState' defines a set of recommended items in previous steps that were liked by the user.

$$s_N = \begin{cases} (R_I + selfAttentionRepresentation(R_I, PositiveState))/2 & \text{if } R_I \text{ Liked by user} \\ mean(PositiveState) & \text{else} \end{cases} \tag{6}$$

Where $R_I$ define the recommended item in this time.

Fig 2 shows the proposed approach to determine the next state of the user. Let $x_2$ be the image that the user recently liked. We assume that $\{x_1, x_3, x_4\}$ are the user's favorite collection of images. The following method is used to calculate the next state of the user (s).

Self-attention is the central mechanism of the Transformer. The purpose of Self-Attention is to measure the dependencies of the components in a sequence with each other in order to have a more accurate perception of the whole sequence. In our work, it is assumed that we

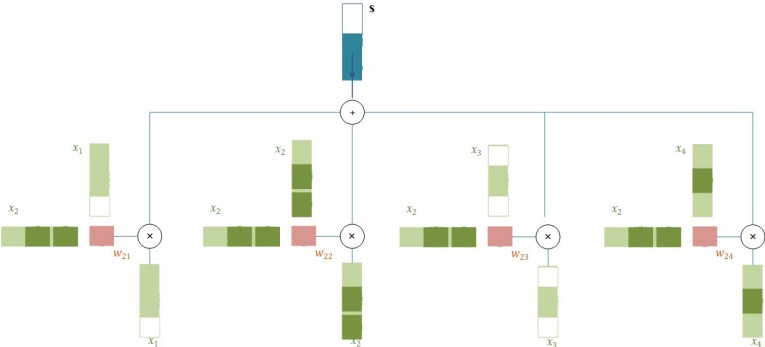

**Fig 2. State representation using self-attention.**

have a sequence of images liked by the user, and the $x_i$ that was recently liked by the user enters the state representation module. To produce the expected output of the entered image (next state of the user), similar to self-attention, a weighted average on the inputs (here, the image that was recently liked is entered, and the liked images from the previous steps are considered as inputs) is calculated.

To generate the output vector *s*, we perform the weighted average operation on all available input vectors:

$$s = \sum_j w_{ij} x_j \tag{7}$$

Where *j* indexes over the whole sequence and the sum of the weights equals one over all *j*, the weight $w_{ij}$ is derived from a function over $x_i$ and $x_j$. The simplest option for this function is the dot product:

$$w'_{ij} = x_i^T x_j \tag{8}$$

Note that $x_i$ is the input vector that was recently liked, and $x_j$ is the image that was previously liked by the user.

The dot product yields a value ranging from negative infinity to positive infinity. To normalize these values to a range of [0,1] and ensure they sum to 1 across the entire sequence, we employ a softmax function.

$$w_{ij} = \frac{exp\ w'_{ij}}{\sum_j \exp\ w'_{ij}} \tag{9}$$

## 4 Experiment

### 4.1 Dataset

In this study, we used four datasets. 1) The Flickr Style dataset [47] for recognizing image style. The original dataset consisted of 80,000 images with style labels, and was classified into 20 labels. We could not collect all the images because some were unlinked from Flickr. In this study, 15 style labels are used, and we have 2900 images for each style. Fig 3 shows seven sample images from this dataset, depicting styles such as serene, melancholy, ethereal, noir, vintage, romantic, and horror. We use this dataset to train a model to extract style features as defined in Table 1.

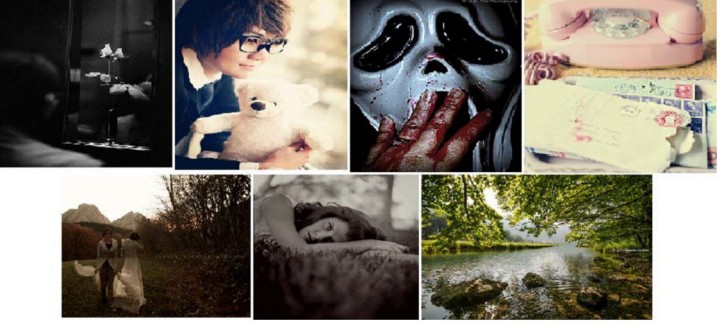

**Fig 3. Random sample images from the Flickr datasets [47].** From left to right, the styles are noir, romantic, horror, and vintage in the first row. From left to right, the styles are ethereal, melancholy, and serene in the second row.

**Table 2. Statistics of the current labeled image data set.**

| Sadness | Fear | Excitement | Disgust | Contentment | Awe | Anger | Amusement | Sum |
|---|---|---|---|---|---|---|---|---|
| 2577 | 969 | 2725 | 1592 | 5130 | 2881 | 1176 | 4724 | **21774** |

2) The data set introduced in [35] was utilized for emotion detection in images. It encompasses eight emotions as defined in Table 2, which include amusement, anger, awe, contentment, disgust, excitement, fear, and sadness. The version of the dataset we used contains fewer images than the original due to some images being lost. Table 2 lists the eight emotions and the corresponding number of images for each, while Fig 4 shows sample images with their labeled emotions.

3) PsychoFlickr dataset is used for personality analysis [40]. A collection of 60,000 images tagged as favorites by 300 Pro Flickr users (200 randomly selected favorites per user). This dataset is used to train a model for extracting Personality features as defined in Table 1.

4) To evaluate image recommendation, we employ part of the dataset used in [1]. We consider a collection of 4000 images belonging to 20 users from Flickr. For each user, there are 200 images tagged as favorites. Random samples of images tagged as favorites by users are shown in Fig 5.

The first three datasets are used respectively to train the methods for style extraction, emotion extraction, and personality analysis. The fourth dataset is employed to evaluate the performance of the proposed recommender system.

## 4.2 Implementation results of feature extraction methods

Before we can provide a recommendation system based on RL, we must first extract the relevant features, such as emotion, personality, and style characteristics. Below, we briefly describe the models implemented and their results in extracting these components.

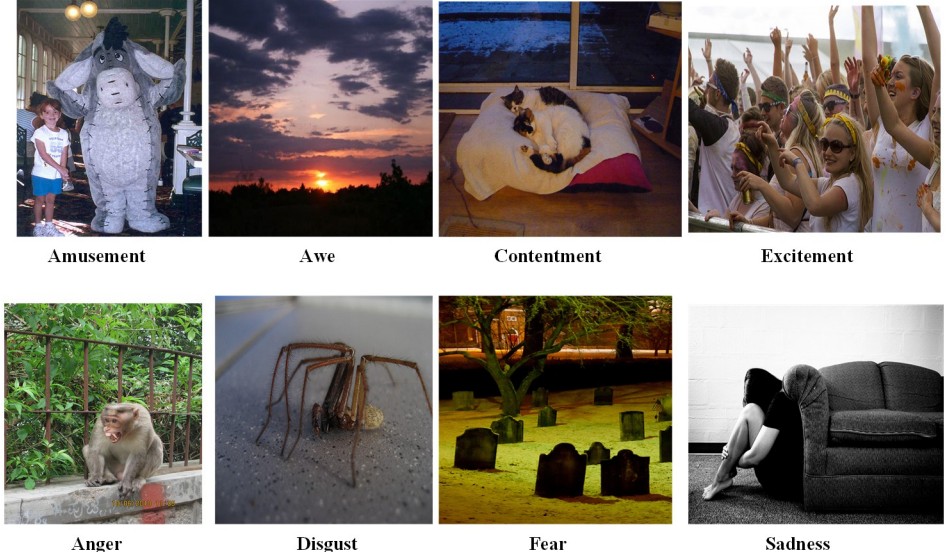

**Fig 4. Sample images of eight different categories of emotions (These images are from the dataset introduced in reference [35]).** Top row: four positive emotions and bottom row: four negative emotions.

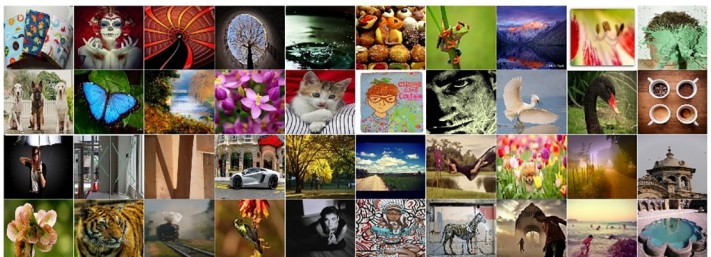

**Fig 5. Random samples of the dataset used for recommendation (These images are from reference [1]).**

**Emotion.** For emotion detection, we developed a system utilizing the EfficientNetB1 model. We began by initializing the network with pre-trained weights from the ImageNet dataset's image classification task. Subsequently, we fine-tuned the network to suit our specific problem. Due to the difference in the number of classes between our dataset and ImageNet, we modified the last fully connected layer to output a probability distribution over the emotional labels of our target dataset. Owing to system limitations, we resized the images to 200×200 pixels.

As detailed in Table 2, the emotion recognition model utilizes a dataset of 21,000 images, which are divided into 80% for training, 15% for testing, and 5% for validation purposes. Table 3 presents the results of the four methods we implemented to recognize emotions. The results indicate that 'FinetuneFMEfficientNetB1 (Imagenet)', which involves the fine-tuning of the EfficientNetB1 network, yields the best performance. Consequently, we have enhanced our recommender system by incorporating emotion analysis with this model.

**Personality.** For personality feature extraction, we have adopted an approach similar to that described in [40]. We attempt to model users' personality traits based on the Five Factor Theory, which involves five binary classifications for each trait. For each trait, the range of values is divided into three parts: the 'low set' includes values below the first quartile, the 'upper set' comprises values above the third quartile, and values in between constitute the 'middle set.' For the binary classification problem, we further refine the distinction between the two classes by selecting only users with values in the low and upper sets.

We have considered two CNN networks, VGG16 and VGG19, which were trained on the ImageNet dataset, and we modified the last layer to adapt to the personality recognition task. These networks are ideal for fine-tuning because they have been trained on a large number of images (1.2 million) across many classes (1000 object categories), providing a robust representational capacity. The results of the implementation are presented in Table 4. For our experiments, we resized the images to 160×160 pixels. We focused on attributed traits since, in the recommender system, we only have information on images liked by the user, and no direct data regarding their personalities.

**Style.** Our previous work [27] introduced a method for image style detection utilizing deep correlation features and a compact convolutional transformer (CCT). This method is based on

**Table 3. The accuracy of different methods for emotion recognition.**

| Method | Accuracy |
|---|---|
| FinetuneFM VGG19 (ImageNet) | 53.49 |
| EfficientNetB1 (ImageNet) | 64.88 |
| EfficientNetB2 (ImageNet) | 64.02 |
| FinetuneFMEfficientNetB1 (ImageNet) | **66.69** |

**Table 4. The accuracy of different methods for personality analysis.**

|  | Attributed traits | | | | |
| --- | --- | --- | --- | --- | --- |
|  | O | C | E | A | N |
| Fine-tune VGG16 | 57.16 | 60.47 | 61.02 | 57.96 | 62.35 |
| Fine-tune VGG19 | 58.4 | 62.08 | 63.79 | 60.87 | 64.33 |

convolution and aims to preserve local information. The key innovation of this method is the proposal of a new convolutional block, replacing the standard block in CCT. For this purpose, we use VGG-19 pre-trained convolutional layers, which were trained using the ImageNet dataset as a convolutional block, and fine-tuned a portion of it during the compression transform learning process. In this paper, we apply the method from[27] to extract image style features.

## 4.3 Implementation results and analysis

In this section, we investigate the effectiveness of the proposed method for a social image recommendation system. For this purpose, we reviewed related works and compared the proposed method with some representative baseline methods. We focused on image-based recommendations and compared our method with others that also centered on image recommendation and do not consider any additional information. To this end, we compared our method with several others that use the same dataset as ours. Additionally, for further investigation, we compared our method with [9] and [48], which use different datasets. We implemented each compared method on our desired dataset to ensure a logical comparison.

We also compared our method with [27] due to their shared emphasis on using style as a feature for recommendation. Although these methods address the problem differently, their comparison is significant as it provides insights into the effectiveness of our method. The proposed method allows for dynamic user interaction with the recommendation system and offers various components to enhance the system. The experimental results have been compared with the results obtained from the following methods:

(i) FGM, this method uses visual features and then applies FGM for simultaneous feature selection and classification [10].

(ii) LASSO, uses visual features and then LASSO regressor [1].

(iii) DeepFeatNN, Deep Features are extracted to learn user preferences. This method generates a user feature vector using averages all the image features liked by the user and then recommends an image that is more similar to the user's vector [9].

(ix) VOFeatFGMPFIS [26], this method use Visual and Object Features. For object features, VGGnet trained on 1000 object from ImageNet tag set [49] was used.

(x) Sulthana el al. [48], this method uses CNN features, applies dimensionality reduction, and employs cosine similarity for image recommendation.

(xi) VisualFeat.FTFMCT. FGM, This method uses style and a visual feature vector and then applies FGM for simultaneous feature selection and classification [27].

To evaluate image recommendation, we consider a dataset comprising 4,000 images from 20 Flickr users, as referenced in[1]. Each user has 200 images tagged as favorites. For comparison purposes, we split the dataset evenly into training and testing sets. Since we only have access to the images liked by users and lack those they did not like, we needed to create two

sets of positive and negative samples for each user. Therefore, for positive samples, we selected 100 images of 200 images "liked" by the user. To determine negative samples, we calculated the similarity between users and then selected 100 images of 200 images "liked" by each of ten users most distance from the target user. For the test set the remaining images of users are used.

In our experiments, we conduct 400 episodes using the proposed method, with 100 images recommended to the user during each episode. We store these 100 images in the recommendation history, whether he likes them or not. At the beginning of each episode, we apply lasso regression on the data stored in the history of recommendations to determine the features that are most important to the user for liking the image. We use the vector obtained from Lasso to filter a large set of images that are candidates for recommendation to the user. Then we use the set of images obtained from this filter to apply Eq (1) and obtain the score of each image. After calculating the scores with Eq (1), the images are ranked based on the scores and the image with the highest score will be recommended. These repetitions and the use of the lasso filter help update the parameters of the actor and critic networks according to the user's interests and advance the system toward providing better recommendations. At the beginning of the system, we set the initial status with one image liked by the user, and at each step, the user's state is updated by the state representation module. This issue can solve the problem of cold start which is one of the problems of recommender systems because the system starts with the least possible information from the user and updates itself with the user's preferences at every step.

After training, for each user, the test images are ranked based on the score given by the recommendation model. We implement different methods and report the top k recommended images, where k is adjustable. We know only the relevant images (user "likes") for each users. We used the following metrics to evaluate recommendation results.

**Recall@k:** this means the proportion of relevant images in the top k. The formulation of Recall@k is as (Eq (10)).

**Precision@k:** means the proportion of relevant top k images, the formulation of Precision@k is as (Eq (11)).

**NDCG@k:** The Normalized Discounted Cumulative Gain (NDCG) measures the relevance of the recommended images and is defined as (Eq 12). Where IDCG@k is the maximum NDCG@k that corresponds to the optimal ranking list so that the perfect NDCGk is 1.

**F1@K:** The F1 Score is a composite metric that incorporates both Precision and Recall using the harmonic mean. In the context of recommendation systems, this F1 Score operates the same way as the regular version (Eq (13)).

$$Recall = \frac{Relevant\_I\,tems\_Recommended\ in\ topk}{Relevant\_I\,tems} \tag{10}$$

$$Precision = \frac{Relevant\_I\,tems\_Recommended\ in\ topk}{k\_Items\_Recommended} \tag{11}$$

$$NDCG@k = \frac{1}{IDCG@k} \times \sum_{i=1}^{k} \frac{G}{\log_2(i+1)} \tag{12}$$

$$F1@K = \frac{2 \times Precision@K \times Recall@K}{Precision@K + Recall@K} \tag{13}$$

In our proposed method, the actor network is composed of two ReLU layers, one Tanh layer, and incorporates a skip connection in the penultimate layer. This network receives the

user state and outputs the action vector, which is then used to make recommendations to the user. The critic network is composed of three ReLU layers, and incorporates a skip connection in the penultimate layer. The dimension of the image embedding vector is determined based on features extracted and is 107. In our experiments, we conduct 400 episodes using the proposed method. In each episode, 100 unique images are recommended to the user; we ensure that items already recommended in previous episodes are not repeated by removing them from the candidate set. Additionally, to make decisions based on the users' most recent interactions, we store up to 100 images (regardless of whether the user liked them or not) recommended in each episode, forming what we refer to as the 'recommendation history'. The discount rate ($\gamma$) is set at 0.3. For all RL-based methods, we employ the Adam optimizer. The mini-batch size is configured at 10, and the initial learning rate is set to 0.001. We determine the optimal parameters, including the hidden layer sizes for both the actor and critic networks, through grid search. In the implementation of our experiments, we utilized Google Colab, a cloud-based platform that provides an accessible and powerful computing environment. Depending on availability, Google Colab typically offers either the NVIDIA K80 or T4 GPUs. The K80 comes with 12GB of GPU memory, whereas the T4 is equipped with 16GB. The virtual machines provided by Google Colab are equipped with 12GB of RAM, ensuring efficient handling of large datasets and computationally intensive tasks. All our code was written in Python. We specifically used Python 3.10.12. Our models were developed and trained using TensorFlow. We employed TensorFlow version 2.12.0, which provided the necessary tools and functionalities for our models.

We calculate the Recall@K and Precision@K on all users using Eqs (2) and (3). Figs 6 and 7 show the results; we can see that the proposed method outperformed the baselines, demonstrating our proposed method's effectiveness. Recall@K is used to calculate the proportion of relevant items in the top-K. Precision@K calculates the proportion of relevant top-K items (liked by the user). Also, in Figs 8 and 9, we have calculated NDCG@k and F1@K for all users and plotted its changes for different k values, and these comparison criteria confirm the superiority of the proposed method.

Fig 10 displays the top 15 recommended images for one of the users, as determined by various methods. We sorted the test images based on the scores from different methods and then selected the 15 highest-scoring images as the top recommendations for each method. The top 15 images recommended to the user by the proposed method are visible in Fig 10h. Notably,

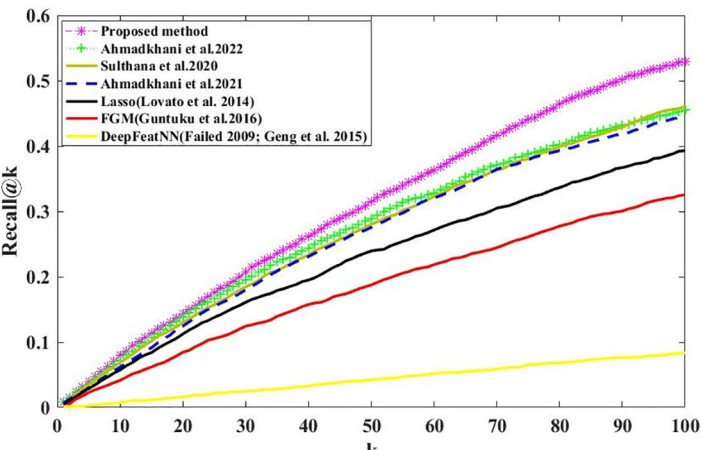

**Fig 6. Recall@k for different k values.**

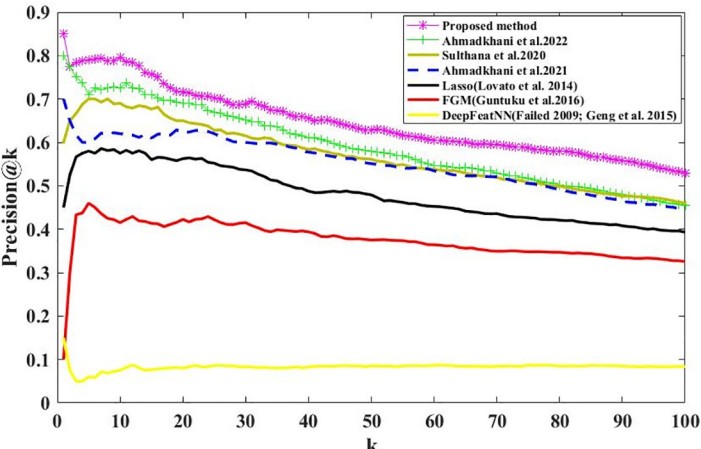

**Fig 7. Precision@k for different k values.**

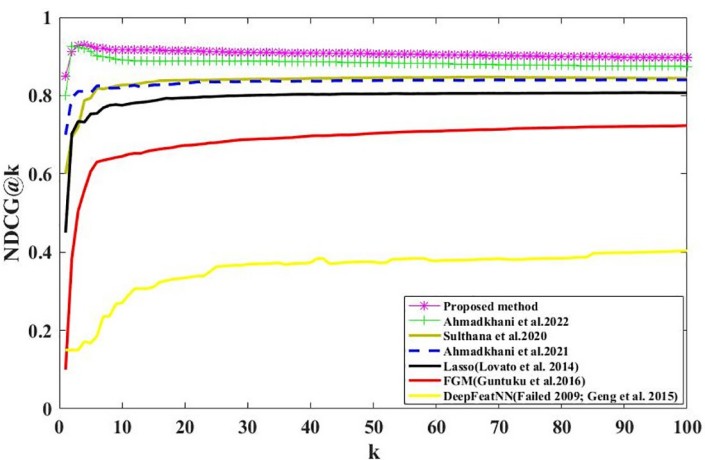

**Fig 8. NDCG@k for different k values.**

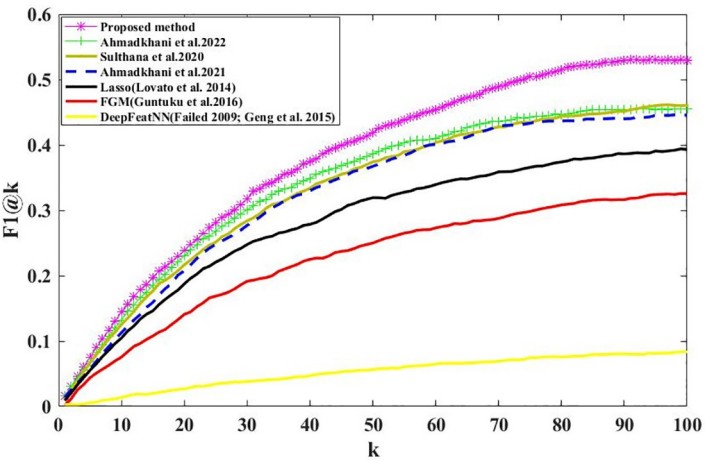

**Fig 9. F1@k for different k values.**

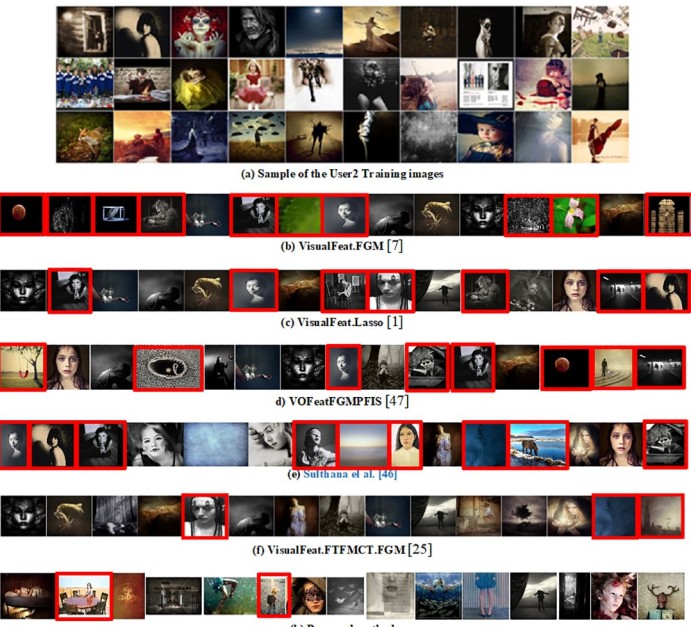

**Fig 10.** (a) some samples of training images for user2. (b)–(h) 15 top images recommended to user2 by different methods. Incorrect recommendations are highlighted in red.

only two of these 15 images were not in the user's collection of favorites. This contrasts with other methods, where a larger number of recommended images did not align with the user's favorites, further demonstrating the proposed method's superiority. We have taken a comprehensive approach by considering the user's preferences from various aspects, analyzing more features such as emotion, style, and personality. Furthermore, by using an interactive approach that updates the system according to the user's preferences, we achieved better results.

Modeling user preferences is a user-centric problem, where the number of images each user likes may be more significant than the total number of users. With this in mind, we chose a set of 200 images per user, in line with the numbers reported in [50] and [51]. Furthermore, we structured our dataset according to the method described in [7], which involves selecting 200 images for each of the 20 users. To validate the results, we expanded the dataset to include 40 users. The results for the two best methods are depicted in Fig 11. As illustrated in Fig 11, the superiority of the proposed method remains evident, even with the increased number of users.

In this paper we proposed a dynamic Image recommender system using deep RL frame work by employing a set of new extracted features includes emotion, style, and personality features. We claimed that using these components in the proposed framework improves the results. Therefore, the ablation study was designed to systematically evaluate the contribution of each component of our proposed framework. By considering one component at a time, we were able to assess its impact on the overall performance. This study has provided valuable insights into how each element contributes to the efficacy of our method. To this end, we considered three approaches:

1. In the first approach, we use visual and style features within the proposed framework to investigate the effect of style features on recommendations (This is shown as (**VS**) in the Figs 12 and 13).

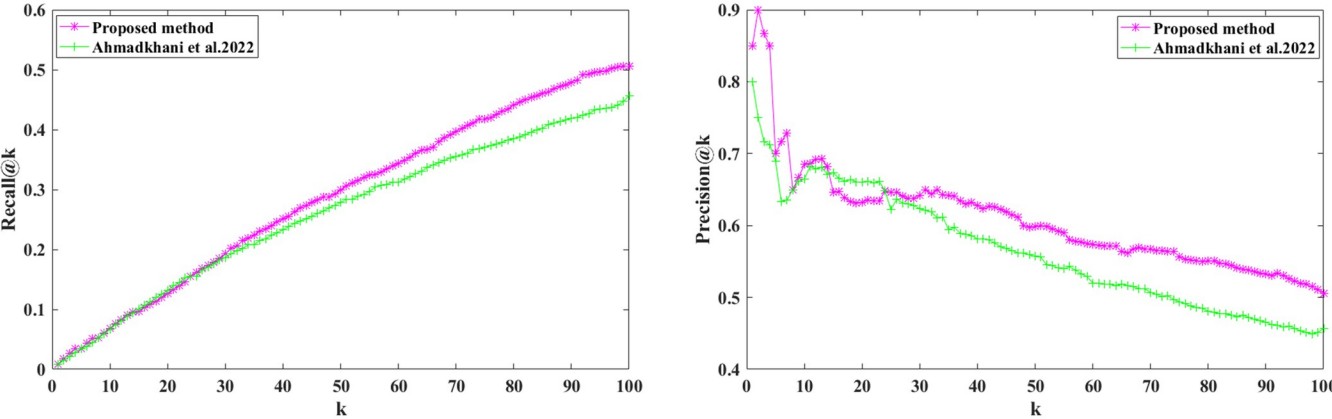

**Fig 11. Recall@k and Precision@k for different k values.**

2. In the second approach, we apply visual and emotion features to examine the impact of emotion features (This is shown as (**VE**) in the Figs 12 and 13).

3. In the third approach, we use visual and personality features within the proposed framework to study the influence of personality features on recommendations (This is shown as (**VP**) in the Figs 12 and 13).

We calculate Recall@K and Precision@K for all users using Eqs (10) and (11) for each approach. The results are shown in Fig 12. The top row displays Recall@K for the three approaches, while the bottom row presents Precision@K. The results underscore the importance of these components within our method. These results indicate that incorporating each of these three components into the proposed framework improves performance. The ablation

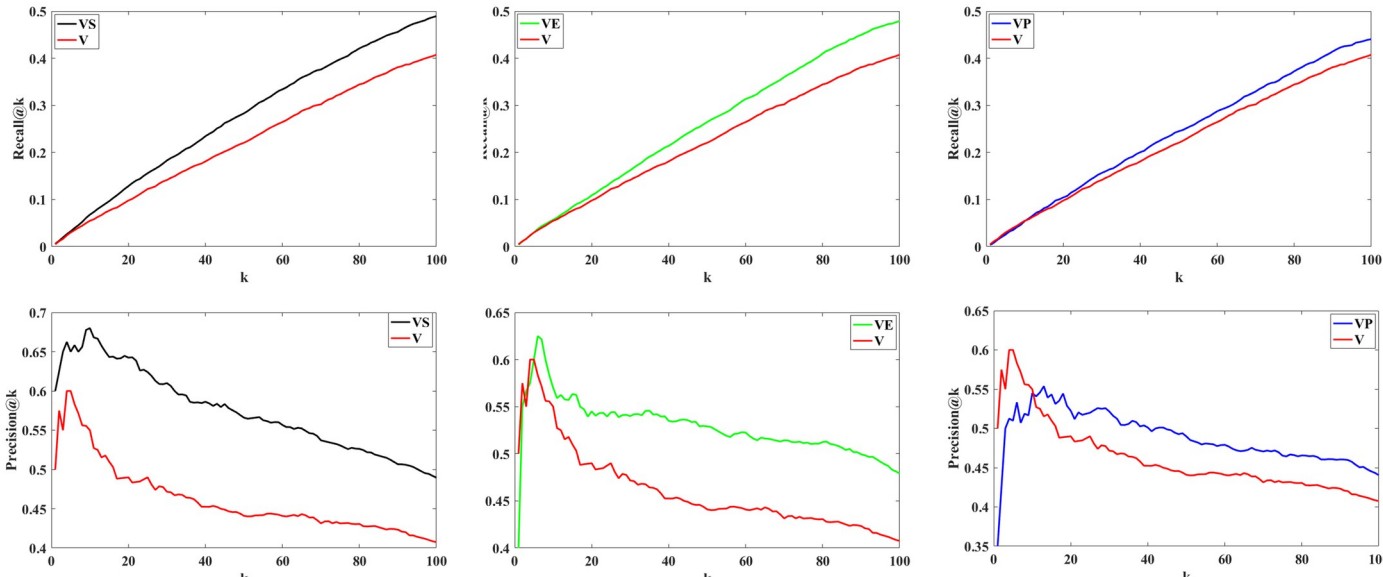

**Fig 12. Shows the results for three approaches: The top row displays Recall@K, and the bottom row shows Precision@K for different k values.**

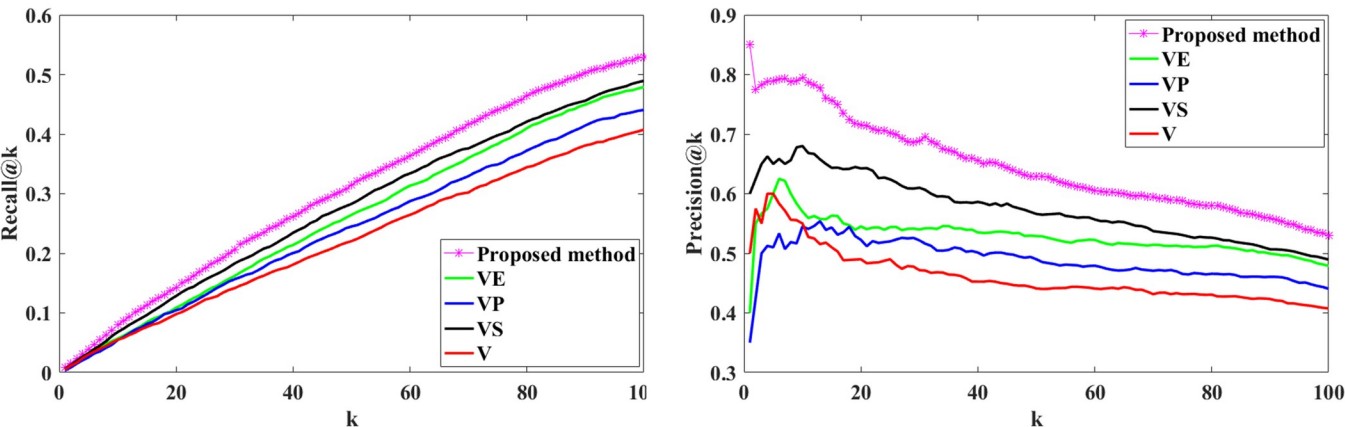

**Fig 13. The comparison between the three approaches and the proposed method.**

study outcomes strongly support our initial hypotheses and assertions about our proposed framework. Additionally, Fig 13 compares the effectiveness of each of these three approaches with the proposed method, which utilizes a combination of all three components within a deep RL framework.

In Table 5, the time consumption of the compared methods is shown. We consider only the training time for the recommendation model and do not consider training time for feature extractor models in the proposed method.

In terms of training time, our method incurs the highest time investment compared to the others. This is due to the additional steps involved in our approach, including the training of a deep learning network. It should be noted that the proposed method interacts with the user, and the training time depends on the number of episodes (The number of episodes use to calculate the time, presented in Table 5, is 100.). Additionally, the proposed method allows us to start learning the user's interests with just one image, even without having initial training data, and gradually refine our understanding of the user's preferences. Therefore, despite the high training time, the advantages outlined for the proposed method signify its beneficial nature.

To calculate the space complexity, we must consider Model Memory and Experience Replay Buffer. For Model Memory, our proposed method maintains two neural networks: the actor (policy) network and the critic (value) network. The space complexity depends on the network architectures and the number of parameters in each network. DDPG uses an experience replay buffer to store past experiences for training. The space complexity of the replay buffer depends on its capacity, which is typically set to hold a fixed number of experiences (e.g., M experiences). The space complexity is O(M). Additionally, we propose considering the

**Table 5. Comparison of running times (in seconds) of different methods.**

| | Training time(s) | Test time(s) |
|---|---|---|
| **FGM(Guntuku et al.2016)** | 1.32 | 0.003 |
| **Lasso(Lovato et al. 2014)** | 1.74 | 0.005 |
| **Sulthana et al.2020** | 1794 | 0.028 |
| **Ahmadkhani et al.2021** | 62 | 0.023 |
| **Ahmadkhani et al.2022** | 2 | 0.01 |
| **Proposed method** | 2323 | 0.05 |

memory for the recommendation history. This memory stores all images recommended to the user, including both liked and disliked images, and is designed to hold a fixed number of experiences (e.g., Z experiences). The space complexity is O(Z).

In the following, we explore the advantages and drawbacks of our proposed framework. User-Centric Design stands out as a significant advantage. Our system comprehensive approach, incorporating user preferences in terms of emotion, style, and personality, represents a significant advancement in personalization. This level of customization ensures that recommendations are more aligned with individual users' tastes, potentially enhancing user satisfaction and engagement. Additionally, Interactive Learning is another advantage. The system's interactive nature, which adapts and updates based on user feedback, is a major strength. This dynamic learning process enables the system to become more accurate over time, tailoring its recommendations more effectively to each user's evolving preferences. Furthermore, the ability to initiate learning without requiring a large dataset from the onset is a noteworthy advantage. This feature makes the system particularly valuable for new users or in applications where historical data is limited, ensuring a seamless and engaging user experience from the start. Continuous Improvement is yet another advantage. The proposed method's capacity to interact with the user from the beginning and gradually improve recommendations demonstrates its potential for real-world applications, where user preferences can vary widely and change over time.

In contrast, the significant training time required due to the model depth and complexity, especially when compared to other methods, could be seen as a drawback. Additionally, the space complexity associated with maintaining dual neural networks, the experience replay buffer, and a memory of recommendation history might pose challenges. While the advanced features and interactive approach are beneficial for personalization, they demand substantial computational resources. Although the system offers a high degree of personalization, there's a trade-off between the model's complexity and its practicality.

In conclusion, our proposed dynamic image recommender system presents a novel and user-centric approach to personalization, leveraging deep reinforcement learning and a broad spectrum of user preferences. While the advantages highlight its potential for highly tailored recommendations and adaptability, the drawbacks show the challenges related to training time, space complexity, and computational demands. Balancing these factors is crucial for maximizing the system's real-world applicability and user satisfaction.

To assess the empirical performance of our algorithm and complement the theoretical time complexity analysis, we executed our code on the Google Colab platform. The environment used for our tests included a runtime equipped with a GPU. The Colab notebooks were configured with a Python environment, and the code was executed using the pre-installed libraries available on Colab at the time of our experiments. It is important to note that although Google Colab environments are standardized in terms of software, the exact hardware specifications can vary and are typically abstracted from the user. Therefore, the reported execution times should be considered as approximations.

## 5. Conclusion

Due to the encounter with the huge volume and variety of information, we need a system that can automatically identify and cater to the user's interests. The remarkable point about previous methods is that they often view the recommendation process as static, assuming that a user's underlying preferences remain unchanged. In reality, personalizing image recommendations according to individual user preferences is a complex task, as their preferences evolve over time. This evolution necessitates dynamic interaction between the system and the user,

indicating the need for a dynamic recommendation process. Additionally, when limited data is available, it becomes difficult for a model to make accurate recommendations, an issue commonly referred to as the cold start problem. In this paper, we propose a dynamic image recommender system using a deep RL framework and introduce a new method for state representation. This framework is novel for social image recommendations. Furthermore, we employ a set of newly extracted features, such as emotion, style, and personality, to enhance efficiency. In Section 4, the results of our tests demonstrate how the use of these components improves outcomes. Overall, the results indicate that our proposed method significantly outperforms some related works in personalized image recommendation. For future work, exploring additional behavioral features related to user behavior is possible. Methods that enhance accuracy in detecting emotions, style, and personality can be investigated, potentially improving the recommender system. We have utilized a combination of manual features and those obtained using deep learning models. Future research could explore other deep learning methods capable of automatically capturing implicit features to achieve better performance. Additionally, self-attention, the core mechanism of the Transformer, focuses on measuring dependencies within a sequence's components to gain a more accurate understanding of the entire sequence. In our work, we employed this concept to calculate the new state. Our approach does not have any trainable parameters, which could be considered in future research.

## Supporting information

**S1 Appendix.**
(DOCX)

## Author Contributions

**Conceptualization:** Somaye Ahmadkhani, Mohsen Ebrahimi Moghaddam.

**Investigation:** Somaye Ahmadkhani, Mohsen Ebrahimi Moghaddam.

**Methodology:** Somaye Ahmadkhani.

**Software:** Somaye Ahmadkhani.

**Supervision:** Mohsen Ebrahimi Moghaddam.

**Validation:** Somaye Ahmadkhani.

**Writing – original draft:** Somaye Ahmadkhani.

**Writing – review & editing:** Mohsen Ebrahimi Moghaddam.

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
