## [Decision Letter · Decision Letter 0]

2 Oct 2023

PONE-D-23-26006A social image recommendation system based on deep reinforcement learningPLOS ONE

Dear Dr. Moghaddam,

Thank you for submitting your manuscript to PLOS ONE. After careful consideration, we feel that it has merit but does not fully meet PLOS ONE’s publication criteria as it currently stands. Therefore, we invite you to submit a revised version of the manuscript that addresses the points raised during the review process.

We look forward to receiving your revised manuscript.

Kind regards,

Nouman Ali

Academic Editor

PLOS ONE

Reviewers' comments:

Reviewer's Responses to Questions

**Comments to the Author**

1. Is the manuscript technically sound, and do the data support the conclusions?

Reviewer #1: Partly

Reviewer #2: Yes

Reviewer #3: Yes

Reviewer #4: Partly

2. Has the statistical analysis been performed appropriately and rigorously? 

Reviewer #1: N/A

Reviewer #2: Yes

Reviewer #3: N/A

Reviewer #4: No

3. Have the authors made all data underlying the findings in their manuscript fully available?

Reviewer #1: Yes

Reviewer #2: Yes

Reviewer #3: Yes

Reviewer #4: Yes

4. Is the manuscript presented in an intelligible fashion and written in standard English?

Reviewer #1: No

Reviewer #2: Yes

Reviewer #3: No

Reviewer #4: No

5. Review Comments to the Author

Reviewer #1: This paper presents a dynamic image recommender system utilizing deep reinforcement learning and incorporating novel features like emotion, style, and personality. Authors introduce a new method for state representation in reinforcement learning to overcome this limitation. Experimental results demonstrate that proposed approach significantly enhances Recall and Precision for personalized image recommendations. Below are some my concern:

• The language used throughout the manuscript needs to be improved

• Improve the introduction part by making it more concise.

• More discussion on the current limitations on machine learning techniques can be helpful.

• Detailed analysis and more results should be added.

• Include in conclusion session more furutre suggestions.

• Include the more comparison metrics.

Reviewer #2: This paper introduced a social image recommendation system through reinforcement learning method. The authors introduce the current development of recommender system and its corresponding related works. Thereafter, they vividly depict the proposed method including numerous components. Then, they conducted some experiments on four datasets comparing with some baselines. Finally, they draw a conclusion that the proposed method makes progress among some previous image recommendation methods.

Innovative points of the proposed method are as follows.

1. They apply pretrained image feature extractors in reinforcement learning based image recommendation tasks.

2. They proposed a new state representation method for image recommendation.

3. Last but not least, the vector representing user interest is applied to better capture user preference.

Some weaknesses also emerge which are as follows.

1. Some formula and equations are presented in the form of pictures (formula (5), formula (6)), which ought to be replaced by latex generated formula.

2. Some images claiming the core structure of the proposed method is presented in the form of bitmap (fig 1. and fig 2.). I don’t know if it is the transfer problem between docx file and pdf file. But in my opinion, such pictures are recommended to be presented in the form of vector graphics. Moreover, the layout of fig 1. is too sparse. Some components of the image lie too far between each other, it’s recommended to be re-arranged.

3. Experiments seems Inadequate. Some parameter/hyper-parameter analysis or ablation study may be conducted to prove the robustness and effectiveness of different components in the whole algorithm procedure.

4. The methods introduced in chapter 3-1) are some manually feature extractors. In the era of deep learning, many DL methods which can automatically capture implicit features, from my point of view, may be tried to get better performance. Manual feature setting should be avoided as much as possible. Such progress may be introduced in future works.

5. The methods introduced in chapter 3-2-1) named ‘self-attention’ is not the ‘self-attention’ in the present sense. It has no trainable parameter. I think a trainable self-attention module is more reasonable. Such progress may be introduced in future works.

To be concluded, the proposed work is considered to be slightly innovated by combining different methods together. But the outcome is proved to be impressive. Some faults exist in the paper which should be removed. The paper needs minor revision.

Reviewer #3: 1) The paper need proofreading and editing to reach standard English.

2) Please refer to the attached file to get my comments!

3)The main contribution(s) of the paper should be expressed clearly.

3)Complexity analysis of the proposed method may be reported at least experimentally.

4) Please add some recent references (I mean after 2022) to the paper.

5)You use many subjects without any justification. If it is possible, please mention the reasons for your decisions where you made to build the proposed method.

Reviewer #4: Today, due to the expansion of the Internet and social networks, people are faced with a huge amount of dynamic information. Recommender systems have emerged in order to improve the problem of the phenomenon of information abundance by analyzing the background of the user's behavior and extracting the user's interests and preferences. Most of the proposed methods for social image recommender systems since now use a non-dynamic strategy, this means they cannot adopt the method by changing user preferences. In this paper, authors have proposed a dynamic Image recommender system using deep reinforcement learning frame work by employing a set of new extracted features such as emotion, style, and personality features. In the proposed method, they overcome the change of state representation definition in reinforcement learning by introducing a new method for state representation. The experimental results show that our proposed method compared to some related works significantly improve Recall@k and Precision@k for personalized image recommendation. There are some novel contributions of this work, author must address my following queries

1. Author must proof read the manuscript to fix grammar errors.

2. Author must present the run-time analysis of proposed research.

3. The sample size use in this search model is quite small, I will suggest the authors to evaluate the proposed research on a large-scale deep learning dataset or otherwise a clear justification is required about this.

4. The comparison with the baseline deep learning research models is missing.

5. The contributions of proposed work are not clear, author must explicitly state the contributions that how each of the claim has been achieved.

6. Author are suggested to cite and discuss

• Nimrah, S., Saifullah, S.: Context-free word importance scores for attacking neural networks. Journal of Computational and Cognitive Engineering.1(4), 187–192 (2022). https://doi.org/10.47852/bonviewJCCE2202406

• Ma, Y., et al.: Deep learning framework for multi-round service bundle recommendation in iterative mashup development. CAAI Trans. Intell. Technol. 1– 17 (2022). https://doi.org/10.1049/cit2.12135

• Fang, Q., et al.: Target-driven visual navigation in indoor scenes using reinforcement learning and imitation learning. CAAI Trans. Intell. Technol. 7( 2), 167– 176 (2022). https://doi.org/10.1049/cit2.12043

7. Results must be compared with state-of-the-art published values.

8. There are many blurry figures that must be redrawn

9. Optimization details are not clear in the current version.

10. Mathematical model is not explained well.

6. PLOS authors have the option to publish the peer review history of their article (what does this mean?). If published, this will include your full peer review and any attached files.

Reviewer #1: No

Reviewer #2: No

Reviewer #3: No

Reviewer #4: No

---

## [Author Response · Author response to Decision Letter 0]

16 Nov 2023

1) Response: 

I reviewed the PLOS ONE style templates thoroughly to ensure that our manuscript complied with all the specified guidelines. Any necessary adjustments, particularly regarding file naming conventions, were diligently implemented to meet the journal's requirements.

2) Response: 

In line with the journal's policy, I am fully prepared to make all author-generated code associated with our manuscript publicly available. Upon publication, the code will be accessible without any restrictions. We will ensure that the code is well-documented, allowing other researchers to understand and reuse it efficiently.

To achieve this, we are pleased to inform you that a portion of our code is already has been made publicly available on GitHub. You can access the code and related materials at the following GitHub repository:

https://github.com/Samadkhani/ImageRecommendation/tree/main

However, we would like to acknowledge that we are actively working on completing the remaining portions and anticipate having them ready for public access in the near future. 

We are open to any additional recommendations or requirements you may have regarding the code sharing process.

3) Response: 

I have an ORCID iD and will ensure that it is validated in the Editorial Manager system as per the guidelines. I will follow the instructions provided in the 'Update my Information' section to fetch/validate my ORCID iD against my account in Editorial Manager.

4)Response: 

To address this concern, we have described the dataset and its location in detail on our GitHub repository, which contains the minimal data set underlying the results described in our manuscript. You can find this dataset and related materials at the following GitHub link:

https://github.com/Samadkhani/ImageRecommendation/tree/main

Our GitHub repository includes comprehensive documentation and access instructions to ensure that the minimal data set is readily available to replicate the study's findings in their entirety.

---

## [Decision Letter · Decision Letter 1]

4 Dec 2023

PONE-D-23-26006R1A social image recommendation system based on deep reinforcement learningPLOS ONE

Dear Dr. Moghaddam,

Thank you for submitting your manuscript to PLOS ONE. After careful consideration, we feel that it has merit but does not fully meet PLOS ONE’s publication criteria as it currently stands. Therefore, we invite you to submit a revised version of the manuscript that addresses the points raised during the review process.

We look forward to receiving your revised manuscript.

Kind regards,

Nouman Ali

Academic Editor

PLOS ONE

Journal Requirements:

Reviewers' comments:

Reviewer's Responses to Questions

**Comments to the Author**

1. If the authors have adequately addressed your comments raised in a previous round of review and you feel that this manuscript is now acceptable for publication, you may indicate that here to bypass the “Comments to the Author” section, enter your conflict of interest statement in the “Confidential to Editor” section, and submit your "Accept" recommendation.

Reviewer #1: All comments have been addressed

Reviewer #2: All comments have been addressed

2. Is the manuscript technically sound, and do the data support the conclusions?

Reviewer #1: Yes

Reviewer #2: Yes

3. Has the statistical analysis been performed appropriately and rigorously? 

Reviewer #1: I Don't Know

Reviewer #2: Yes

4. Have the authors made all data underlying the findings in their manuscript fully available?

Reviewer #1: Yes

Reviewer #2: Yes

5. Is the manuscript presented in an intelligible fashion and written in standard English?

Reviewer #1: Yes

Reviewer #2: Yes

6. Review Comments to the Author

Reviewer #1: none, no further suggestion, all comment addressed ------------------------------------------------------------------------------------------------

Reviewer #2: In this paper, the authors introduce a framework based on actor-critic reinforcement learning with a state representation module to a dynamic image recommendation system. Features like emotion, style, and personality are employed to enhance adaptability and comprehensive experiments are conducted on public datasets to demonstrate the effectiveness. Although some valuable ideas and explorations are discussed in the paper, there are still several areas that require explanation and modification. Here are some suggestions for the authors to improve the article.

Overall Comment: Minor Revisions

1. The authors are advised to thoroughly review the manuscript as inconsistencies in font sizes have been observed in the text. A notable example of this inconsistency can be found in Lines 148-158.

2. Please do not add arXiv preprint papers to the reference list, as they are not peer-reviewed. Replace them with other similar papers.

3. In the related works section, authors should compare the differences and shortcomings of different works rather than simply listing and summarizing them.

4. The specific terms in the manuscript lack consistency. For instance, "deep reinforcement learning" is abbreviated as "DRL" in the Introduction, but variations such as "Deep RL" and "Deep Reinforcement Learning" persist in subsequent chapters.

5. The quality of the figures in this paper can be greatly improved. In the current version of this paper, some figures have small font sizes, which are hard to read, such as Fig. 1.

6. The authors should explain the models trained deeply, showing more details of the experiments, such as the experiment running environment, including software and hardware configurations.

7. PLOS authors have the option to publish the peer review history of their article (what does this mean?). If published, this will include your full peer review and any attached files.

Reviewer #1: No

Reviewer #2: No

---

## [Author Response · Author response to Decision Letter 1]

9 Dec 2023

Thank you for your valuable feedback regarding the reference list. We have meticulously reviewed our reference list to ensure its completeness and accuracy. We understand the importance of relying on credible and current sources, and we replaced these references with the most relevant and recent research in our field.

In light of reviewer advice, we have reviewed our reference list and replaced the arXiv preprint citations with alternative sources. We have ensured that these replacements are from peer-reviewed journals or conferences, closely related to the subject matter of the removed references, and equally contribute to the context and support of our research findings. References 14, 29, and 30 have been changed.

---

## [Decision Letter · Decision Letter 2]

2 Feb 2024

PONE-D-23-26006R2A social image recommendation system based on deep reinforcement learningPLOS ONE

Dear Dr. Moghaddam,

Thank you for submitting your manuscript to PLOS ONE. After careful consideration, we feel that it has merit but does not fully meet PLOS ONE’s publication criteria as it currently stands. Therefore, we invite you to submit a revised version of the manuscript that addresses the points raised during the review process.

We look forward to receiving your revised manuscript.

Kind regards,

Nouman Ali

Academic Editor

PLOS ONE

Journal Requirements:

Reviewers' comments:

Reviewer's Responses to Questions

**Comments to the Author**

1. If the authors have adequately addressed your comments raised in a previous round of review and you feel that this manuscript is now acceptable for publication, you may indicate that here to bypass the “Comments to the Author” section, enter your conflict of interest statement in the “Confidential to Editor” section, and submit your "Accept" recommendation.

Reviewer #1: All comments have been addressed

2. Is the manuscript technically sound, and do the data support the conclusions?

Reviewer #1: Partly

3. Has the statistical analysis been performed appropriately and rigorously? 

Reviewer #1: N/A

4. Have the authors made all data underlying the findings in their manuscript fully available?

Reviewer #1: Yes

5. Is the manuscript presented in an intelligible fashion and written in standard English?

Reviewer #1: Yes

6. Review Comments to the Author

Reviewer #1: • Authors need to add a table of used symbols in the paper to make the paper read easier.

• What specific improvements the authors consider regarding the methodology?

• Language of the paper is too poor. Need a rigorous look. In many places few words are repeatedly used in one sentence or paragraph. That degrades the readability of the paper. Rewrite abstract.

• What is your fitness function? what is the outcome of the proposed algorithm?

• Conduct a more thorough comparison with state-of-the-art algorithms, such as Hybrid Feature Selection Techniques Utilizing Soft Computing Methods for Cancer Data, Gene selection with Game Shapley Harris hawks optimizer for cancer classification, CO‐WOA: Novel Optimization Approach for Deep Learning Classification of Fish Image.

• Provide a balanced discussion on the advantages and drawbacks of the proposed work.

7. PLOS authors have the option to publish the peer review history of their article (what does this mean?). If published, this will include your full peer review and any attached files.

Reviewer #1: No

---

## [Author Response · Author response to Decision Letter 2]

15 Feb 2024

According to the suggestion of the Reviewer, the following three articles have been added to the list of references:

6. Aziz, R.M., et al., Hybrid Feature Selection Techniques Utilizing Soft Computing Methods for Cancer Data, in Computational and Analytic Methods in Biological Sciences. 2023, River Publishers. p. 23-39.

7. Aziz, R.M., et al., CO‐WOA: novel optimization approach for deep learning classification of fish image. Chemistry & Biodiversity, 2023. 20(8): p. e202201123.

8. Afreen, S., A.K. Bhurjee, and R.M. Aziz, Gene selection with Game Shapley Harris hawks optimizer for cancer classification. Chemometrics and Intelligent Laboratory Systems, 2023. 242: p. 104989.

---

## [Decision Letter · Decision Letter 3]

21 Feb 2024

A social image recommendation system based on deep reinforcement learning

PONE-D-23-26006R3

Dear Dr. Moghaddam,

We’re pleased to inform you that your manuscript has been judged scientifically suitable for publication and will be formally accepted for publication once it meets all outstanding technical requirements.

Kind regards,

Nouman Ali

Academic Editor

PLOS ONE

Additional Editor Comments (optional):

Reviewers' comments:

Reviewer's Responses to Questions

**Comments to the Author**

1. If the authors have adequately addressed your comments raised in a previous round of review and you feel that this manuscript is now acceptable for publication, you may indicate that here to bypass the “Comments to the Author” section, enter your conflict of interest statement in the “Confidential to Editor” section, and submit your "Accept" recommendation.

Reviewer #1: All comments have been addressed

2. Is the manuscript technically sound, and do the data support the conclusions?

Reviewer #1: Yes

3. Has the statistical analysis been performed appropriately and rigorously? 

Reviewer #1: N/A

4. Have the authors made all data underlying the findings in their manuscript fully available?

Reviewer #1: Yes

5. Is the manuscript presented in an intelligible fashion and written in standard English?

Reviewer #1: Yes

6. Review Comments to the Author

Reviewer #1: None, all suggested comments have been incorporated in the revised version. The revised manuscript now meets the journal's requirements.

7. PLOS authors have the option to publish the peer review history of their article (what does this mean?). If published, this will include your full peer review and any attached files.

Reviewer #1: No

---

## [Editor Report · Acceptance letter]

26 Mar 2024

PONE-D-23-26006R3 

PLOS ONE

Dear Dr. Moghaddam, 

I'm pleased to inform you that your manuscript has been deemed suitable for publication in PLOS ONE. Congratulations! Your manuscript is now being handed over to our production team.

Kind regards, 

on behalf of

Dr. Nouman Ali 

Academic Editor

PLOS ONE